# High coverage and equitable distribution of COVID-19 vaccine uptake in two vulnerable areas in Bangladesh

**Muhammed Nazmul Islam**[1*], **Manuela De Allegri**[2], **Emmanuel Bonnet**[3], **Malabika Sarker**[1], **Jean-Marc Goudet**[4], **Lucas Franceschin**[4], **Valéry Ridde**[4, 5]

1 BRAC James P Grant School of Public Health, BRAC University, Dhaka, Bangladesh, 2 Heidelberg Institute of Global Health, University Hospital and Medical Faculty, University of Heidelberg, Heidelberg, Germany, 3 IRD UMR 215 PRODIG, CNRS Université Paris 1 Panthéon-Sorbonne, AgroParisTech, Aubervilliers, France, 4 CEPED, IRD-Université de Paris, ERL INSERM SAGESUD, Paris, France, 5 French Collaborative Institute on Migration, Paris, France

* nazmulislam@sajida.org

## Abstract

Bangladesh completed a primary series of COVID-19 vaccinations for about 86 individuals per 100 population as of 5 July 2023. However, ensuring higher coverage in vulnerable areas is challenging. We report on the COVID-19 vaccine uptake and associated factors among adults in two vulnerable areas in Bangladesh. We conducted a cross-sectional study between August and September 2022 in Duaripara, a slum in northeast Dhaka (in-migration site), and Tala, a disaster-prone sub-district in southwest Satkhira (out-migration site). We surveyed 1,239 adults in Duaripara and 1,263 adults in Tala from 625 and 596 randomly selected households, respectively. We reported coverage and examined associations between the uptake and demographic and socioeconomic characteristics using multilevel mixed-effects generalized linear regression models. We checked for spatial autocorrelation to assess geographical patterns in vaccine distribution. First- and second-dose coverage was about 91% and 80.4% in Duaripara and 96.6% and 92.2% in Tala, respectively. Individuals above 40 were more likely to be vaccinated (IRR: 1.12, p-value = 0.04 for Duaripara, and IRR: 1.14, p-value <0.01 for Tala). Professions requiring more outdoor interactions had a higher likelihood of receiving the vaccine. In Tala, television access (IRR: 2.09, p-value <0.01) and micro-credit membership (IRR: 1.50, p-value = 0.05) were positively associated with receiving a booster dose and negatively associated with smart-phone access (IRR: 0.58, p-value = 0.03). Moreover, temporarily migrated respondents were more likely to be unvaccinated (IRR: 0.87, p-value = 0.04). Income was not associated, indicating equitable distribution. Moreover, no geographical clustering was detected. The credit for high COVID-19 vaccine coverage in Bangladesh can be attributed to the country's longstanding success in implementing immunization programs, which relied on community mobilization and effective health education to generate demand. However, to ensure comprehensive coverage in vulnerable areas, targeted interventions can help increase uptake by addressing specific sociodemographic differences.

**Data availability statement:** All the relevant data are within the paper and its Supporting Information files.

**Funding:** The study was funded by the French National Research Agency (ANR) as part of the presidential call "Make Our Planet Great Again" (Grant number: ANR-18-MPGA-0010 to VR). The funder had no role in study design, data collection and analysis, decision to publish, or preparation of the manuscript.

**Competing interests:** The authors have declared that no competing interests exists.

## Introduction

The COVID-19 pandemic, caused by the SARS-CoV-2 (novel coronavirus), has affected millions worldwide and posed unprecedented challenges for public health and social systems. Asia has been one of the most affected regions, with more than 100 million confirmed cases and over 1.5 million deaths as of November 2022 [1]. Bangladesh, a densely populated country in South Asia, has reported over 2 million cases and over 29,000 deaths since the start of the pandemic [1]. After all the non-pharmaceutical interventions implemented during the pandemic, the effectiveness of which has been widely debated [2], immunization is considered the primary strategy to control the pandemic and prevent its consequences.

While it is true that vaccination against COVID-19 cannot solve all the challenges posed by the pandemic [3], it plays a vital role in reducing the risk of infection, transmission, hospitalization, and death [4]. However, the acceptability and uptake of COVID-19 vaccines vary across countries and populations, depending on supply, communication, trust, knowledge, attitudes, and behaviors [5]. A study on COVID-19 vaccine acceptance in Bangladesh conducted in 2020 revealed that 60.5% of respondents were willing to receive a COVID-19 vaccine if available and that age, gender, education, occupation, income, religion, and the perceived severity of COVID-19 were significant predictors of vaccine acceptance [6]. While the issue of ensuring equity is complex regarding vaccination, a recent meta-analysis confirmed that inequalities in vaccination coverage persist at the global level [7]. Hence, concerns about ensuring equity in vaccine access and distribution remained [8] while the world embarked on a vaccination strategy against COVID-19. Constant worries prevailed about an inverse care law in global vaccination strategy [9] that could potentially leave the poorest or most remote populations overlooked in favor of more privileged population groups. One of the first studies on COVID-19 vaccination in Israel confirmed the presence of this socio-economic gradient [10]. However, empirical studies on inequalities and vaccination against SARS-CoV-2 remain scarce [11], particularly on a local scale and in vulnerable areas.

Bangladesh launched its COVID-19 vaccination program on January 27, 2021, with the aim of immunizing at least 80% of its population by the end of 2022 [12]. The program initially targeted frontline health workers and other high-risk groups. Then, it was gradually expanded to include the population aged 18 years and above from February 7, 2021. The vaccination was provided free of charge [see detail here 12]. As of November 13th, 2022, about 92% of the eligible population had received at least one dose of a COVID-19 vaccine, and about 86% had completed their primary series of vaccinations (i.e., two doses) [9]. These figures indicate a remarkable achievement for a low- and middle-income country (LMIC) facing multiple challenges, such as limited resources, vaccine hesitancy, misinformation, and socioeconomic inequality [12,14]. We applied the TIDieR-PHP framework [13] to describe Bangladesh's COVID-19 vaccination program and its context (see S1 Text).

The high COVID-19 vaccine coverage in Bangladesh can be attributed to the country's longstanding commitment to immunization, with roots dating back to its Expanded Program on Immunization (EPI) launch in the 1970s [14,15]. Bangladesh has gained worldwide recognition for the effectiveness of its immunization programs[15], especially in South East Asia [16]. The keys to success were a pluralistic health system, major outreach approaches, and strong community involvement [17]. According to the latest 2017 Demographic and Health Survey (DHS), the full immunization coverage in Bangladesh was 86% for children aged between 12 and 23 months [18]. However, a study from 2014 revealed that children from the wealthiest families were twice as likely to be fully immunized compared with others, and children residing in urban areas were 1.35 times more likely to be fully immunized than those living in rural areas [17]. Ensuring equity in vaccine distribution still remains an important

concern in Bangladesh [19]. Therefore, in the context of the quick creation and rapid implementation of a national program for COVID-19 vaccination, it is interesting to enquire how Bangladesh upheld its legacy of high vaccination coverage and whether the equity concerns were addressed, especially among populations residing in vulnerable areas [15]. A critical analysis of the country's experience can offer important lessons for public health practice at the global level [20].

This paper presents the coverage of the COVID-19 vaccine and demographic and socio-economic factors associated with the vaccine uptake in two vulnerable areas in Bangladesh: a flood- and waterlogging-prone district in southwest Bangladesh and an urban slum in Dhaka city.

## Methods

### Study sites

We purposively selected an urban and a rural site. Fig 1 illustrates the designated areas on a map. Duaripara was selected as the urban site. It is a densely populated slum located in the peripheries of north-eastern Dhaka (i.e., the capital of Bangladesh). It covers an area of about 0.179 square kilometres and comprises both temporary and permanent settlements (Fig 2). In contrast, we selected Tala as the rural site. It is a disaster-prone sub-district of Satkhira located in the southwest region of Bangladesh. It has a total area of about 344.2 square kilometres, administratively divided into 12 unions (see Fig 3). Despite the differences, both sites were considered vulnerable since the prevalence of climate-induced migration was high in both sites. People from different districts were moving to Duaripara; hence, it was regarded as a high in-migration site. At the same time, Tala was experiencing a high out-migration (i.e., people were moving out to different districts).

### Sampling and data collection

This study was part of an exploratory research program to understand formal healthcare access among people residing in areas vulnerable to climate change [22]. Thus, access to formal healthcare services was used as the variable of interest to estimate the sample size [23]. We defined *formal healthcare service* as receiving healthcare from any recommended primary, secondary, or tertiary healthcare facilities in Bangladesh, which ranges from a community clinic to any specialized hospital. The population of interest included households with (1) any member who had suffered or was suffering from any illness that started within 30 days from the date of the survey, or (2) a pregnant woman, or (3) a mother of any child under two years of age. Since no literature estimated formal healthcare access in Duaripara and Tala, we used estimates from our pilot study to calculate the sample sizes.

The pilot survey found that access to formal healthcare services was about 23% in the Duaripara slum. Therefore, using a 7% precision and a 5% significance level, the estimated sample size for the Duaripara slum was about 555 households. However, the people residing in slums are comparatively mobile. Hence, we estimated the sample size by adjusting for a 20% non-response rate. The final sample size for the Duaripara slum was 694 households.

In contrast, the pilot survey found that access to formal healthcare services was about 19% in Tala. Using a 7.5% precision and a 5% significance level, the estimated sample size for Tala was about 420 households. However, unlike Duaripara, Tala covers a large area; hence, we planned to select households from 10 pre-defined clusters, where a cluster is defined as an area within a 2–3 kilometre radius from the randomly selected community clinic. Assuming an intra-cluster correlation coefficient of 0.01, we adjusted the sample size for Tala by a design

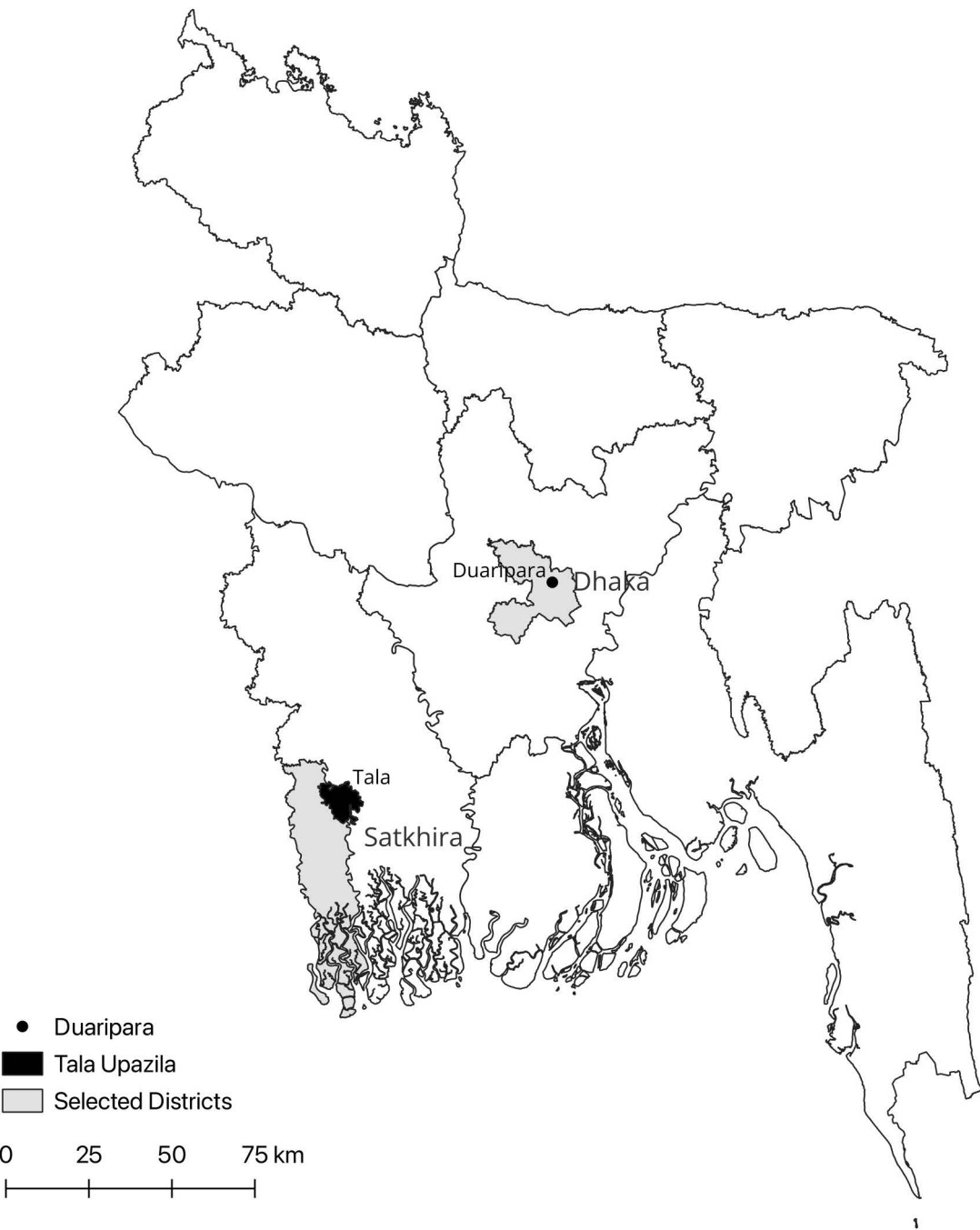

**Fig 1. Study sites.** The map of study sites was plotted in QGIS using shapefiles published by the Humanitarian Data Exchange (HDX) [21]. The shapefiles are publicly accessible.

effect of 1.41 and a 10% non-response rate. The final sample size for Tala was 658 households (see S1 Dataset in the Supporting Information).

We collected the data carrying out a cross-sectional survey from 17 August 2022 to 9 September 2022 (see the STROBE statement in the Supporting Information, S1 Checklist). First, we conducted a listing survey and identified 1,435 eligible households from Duaripara and

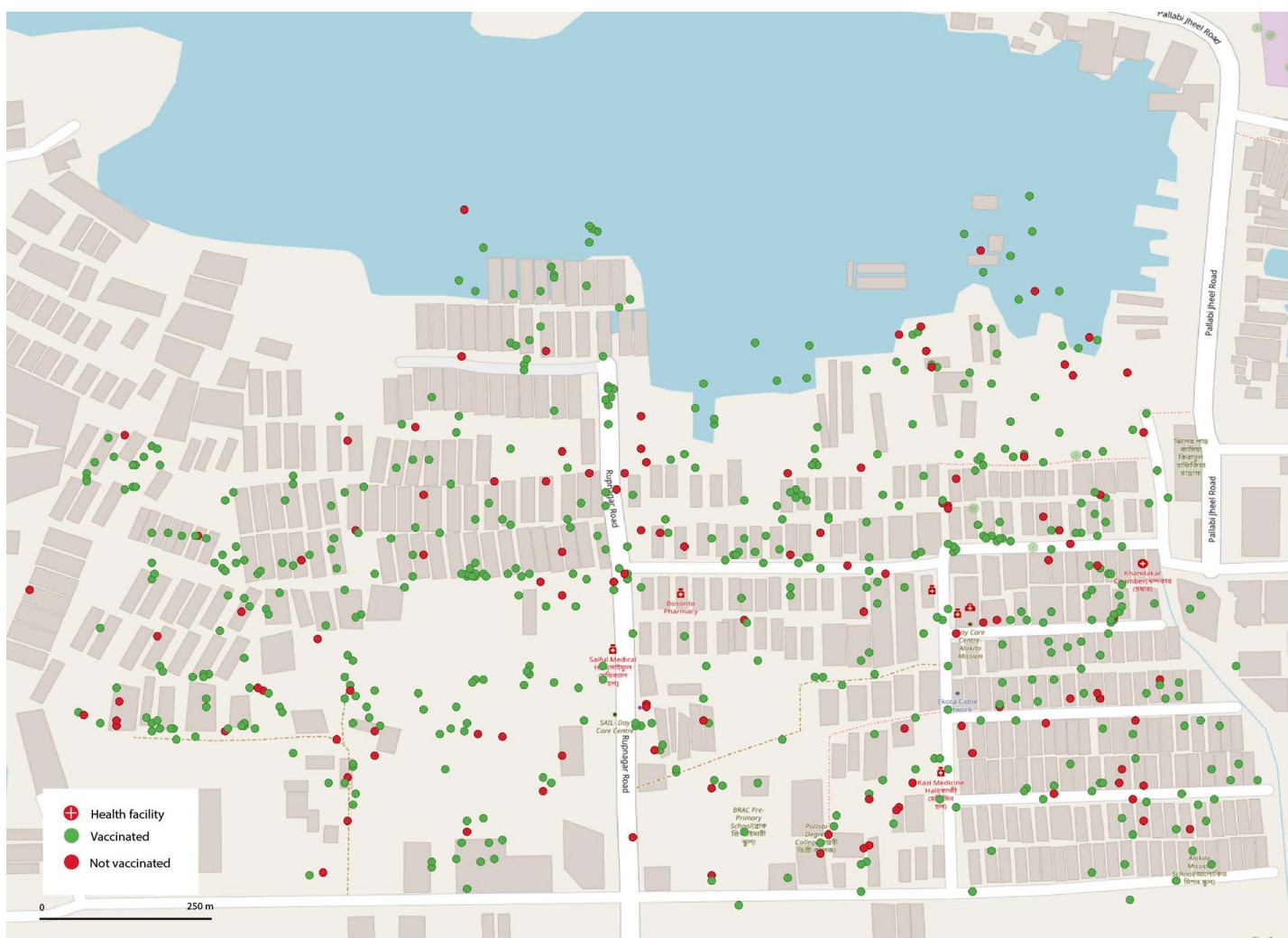

**Fig 2. Vaccinated and unvaccinated respondents in Duaripara.** The map of Duaripara was plotted in QGIS using shapefiles published by the Humanitarian Data Exchange (HDX) [21]. The shapefiles are publicly accessible. During the survey, we collected GPS coordinates of the households where the respondents were resided. Respecting the de-identification policies, we are not sharing the GPS coordinates outside the research team.

2,919 from the ten selected clusters of Tala meeting the inclusion criteria. Then, we selected the study households from this list, applying a simple random sampling technique. Finally, we conducted the survey with 625 households in Duaripara and 596 households in Tala. From each household, Referenced on availability and informed written consent, we targeted to survey four respondents; an adult male and an adult female from each of the two age groups: (1) 18–59 years, (2) 60 years or above.

Moreover, there were a few specific questions for all under-5 children and pregnant women. If the questions were related to a minor (i.e., less than 18 years old), we interviewed their parents or immediate caregivers. We interviewed a total of 1,239 participants from Tala and 1,263 participants from Duaripara. A team of 35 trained surveyors collected the data using SurveyCTO version 2.70.

The survey collected information on participants' demographic and socioeconomic characteristics, such as migration status, history of experiencing climatic events, COVID-19

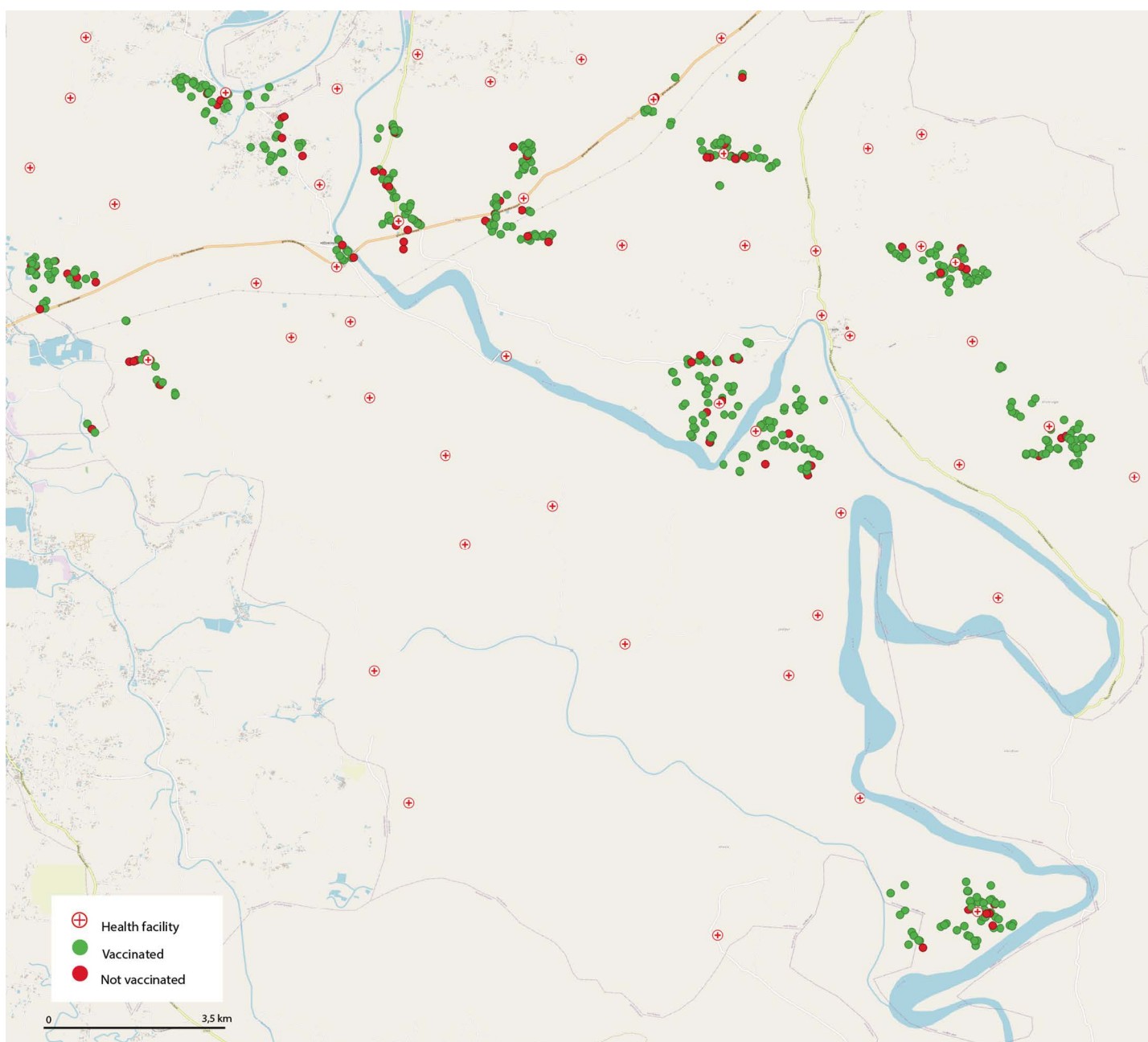

**Fig 3. Vaccinated and unvaccinated respondents in Tala.** The map of Tala was plotted in QGIS using shapefiles published by the Humanitarian Data Exchange (HDX) [21]. The shapefiles are publicly accessible. During the survey, we collected GPS coordinates of the households where the respondents were resided. Respecting the de-identification policies, we are not sharing the GPS coordinates outside the research team.

infection and vaccination status, acute and chronic illness status, healthcare utilization and expenditure, social capital, and mental health. For this study, we used the data on COVID-19 vaccination among adults (i.e., 18 or above years old), their migration statuses, and socio-demographic features since our focus was on assessing COVID-19 vaccination status and factors associated with its uptake.

## Variables and their measurements

Table 1 presents all the variables and how they were measured. We defined three different outcome variables. First, we explored variations in the actual number of COVID-19 vaccine doses received and defined the first outcome variable as a count variable, with 0 quantifying having received no vaccine shot. Second, we differentiated participants who had received at least a full vaccination course of two doses (coded as 1) from those who had not received at least two doses (coded as 0). Finally, in our third outcome variable, we differentiated participants who had received the full course of two doses plus a booster dose (coded as 1) from all others.

We also present the explanatory variables and their hypothesized direction of association with the vaccine uptake (see column 3 of Table 1). Age, gender, marital status, education, and occupation were included as socio-demographic features. Age was initially collected as

**Table 1. Variables, their measurement, and hypothesised direction of the coefficient.**

| Variable | Measurement | Hypothesized direction of the coefficient |
|---|---|---|
| **Outcome variables** | | |
| **Count** | | |
| No. of doses of COVID-19 vaccine | Counting the number of COVID-19 vaccine shots taken by the participant; 0 if not taken the vaccine | |
| **Binary** | | |
| Taken at least two doses of COVID-19 vaccine | 0 = No | |
| | 1 = Yes | |
| Taken two- and a booster-dose of COVID-19 vaccine | 0 = No | |
| | 1 = Yes | |
| **Explanatory variables** | | |
| **Binary** | | |
| Age | 0 = 18–40 years | + |
| | 1 = Above 40 | |
| Gender | 0 = Male | + |
| | 1 = Female | |
| Marital status | 0 = Other | − |
| | 1 = Currently married | |
| Television ownership | 0 = Does not own a TV | − |
| | 1 = Owns a TV | |
| Smart-phone ownership | 0 = Does not own a smartphone | − |
| | 1 = Owns a smartphone | |
| Micro-credit membership | 0 = Not a member | − |
| | 1 = Is a member | |
| Chronic illness status | 0 = Not reported any chronic illness | − |
| | 1 = Suffering from a chronic illness | |
| COVID-19 test | 0 = Not taken any COVID-19 test | − |
| | 1 = Taken a COVID-19 test | |
| Migration status | 0 = Have not stayed outside of home for 30 days or more within the last 12 months | + |
| | 1 = Migrated or stayed outside of home for 30 days or more within the last 12 months | |

*(Continued)*

**Table 1.** (Continued)

| Variable | Measurement | Hypothesized direction of the coefficient |
|---|---|---|
| **Continuous** | | |
| Household size | No. of household members | − |
| Monthly income | Monthly household income in BDT. | + |
| Mental wellbeing score | Score of WHO-5 wellbeing index | + |
| Distance from nearest healthcare facility | In meter | − |
| **Categorical** | | |
| Education | 0 = No education | − |
| | 1 = Primary or less | +/− |
| | 2 = Above primary | + |
| Occupation | 0 = Others | − |
| | 1 = Agriculture | +/− |
| | 2 = Business or self-employed | + |
| | 3 = Service | + |
| | 4 = Day labor | +/− |
| Relationship with household-head | 0 = Household-head | + |
| | 1 = Spouse | − |
| | 2 = Other | +/− |

a continuous variable. However, we dichotomised it (18–40 years and above 40) to check whether the government policy to prioritize elders was reflected in vaccine uptake. Hence, the direction of association between age and vaccine uptake was hypothesized to be positive. We included a categorical variable presenting participants' relationships with their respective household heads. We hypothesized that a household head would have a higher chance of getting vaccinated than any other household member. Definitions of all the socio-demographics are presented in Table 1.

We included monthly household income as an indicator of wealth. We used participants' access to television, access to smartphones, and micro-credit membership status. Access to television and smartphones, and connection with micro-credit institutions indicates that the participant had access to different sources of information. These sources could help get information about COVID-19 and its preventive practices (e.g., vaccines). We hypothesized that access to television or social media could positively affect vaccine uptake. Holding membership in a micro-credit organization is generally related to higher social capital [24,25]. Hence, we also hypothesized this to be positively associated with vaccine uptake.

As proxies for health, we included participants' chronic illness statuses and mental wellbeing scores as covariates. Chronic illness status was a binary variable indicating whether the participant had suffered from any chronic illness during the survey. Given the potential side effects associated with COVID-19 vaccines, individuals with chronic illnesses may exhibit increased hesitancy, which could be negatively associated with vaccine uptake. At the same time, the mental wellbeing score was the WHO-5 wellbeing index that measured participants' wellbeing using a five-item questionnaire. The aggregated score ranged from 0 to 100, where a lower score represents a lower mental well-being. We hypothesized a positive association between mental wellbeing and vaccine uptake.

We included migration as a covariate. It was a binary variable showing whether a participant had stayed outside their home for 30 days or more within the last 12 months. Migration was hypothesized to be negatively associated with vaccine uptake. Finally, for Tala, we included the geographical distance (in meters) between the participants' households and the nearest healthcare facility (i.e., community clinic) as a covariate. However, we did not use distance as a covariate for Duaripara since the urban healthcare system in Bangladesh is much more pluralistic than its rural counterparts. The distance was hypothesized to be negatively associated with the vaccine uptake (see column 3 of Table 1).

## Statistical analysis

We analysed the data separately for the two sites since the sites are entirely different regarding their demographics and living conditions. Duaripara is a densely populated urban slum with poor living conditions and comparatively mobile households; whereas in Tala, households are mostly permanent. However, it is a hard-to-reach rural area frequently affected by natural calamities. Moreover, the local administrations responsible for distributing the COVID-19 vaccine in these two sites were also different and applied different strategies to distribute the vaccine. Duaripara introduced temporary booths where people registered and received the COVID-19 vaccine on the spot. Tala used a pre-registration system where people had to register and visit a nearby facility to get the vaccine.

First, we conducted a descriptive analysis of COVID-19 vaccine uptake for the three doses (i.e., two doses and a booster) and the participants' socio-demographic and economic characteristics (see Table 2 and Table 3). We reported frequency and percentage for binary variables and mean and standard deviation for continuous variables. We also checked for differences in vaccine uptake by explanatory variables using the Chi-squared test of independence for binary/categorical variables and the t-test of difference in means for continuous variables. The results are reported in S1 Table and S2 Table in the Supporting Information.

Second, we used regression models to estimate the association between uptake and the participants' socio-demographic and economic characteristics. Since we recruited multiple participants from the same household and the households were nested in villages, we fitted Multilevel Generalized Linear Mixed-effects Models. For Duaripara, the data structure was hierarchical representing two levels – individuals were nested in households. Therefore, the models took the following functional form:

$$g\left\{E\left(Y_{ij}|\boldsymbol{X}_{ij},c_j\right)\right\} = \beta_0 + \boldsymbol{\beta_x}\boldsymbol{X}_{ij} + c_j \tag{1}$$

where $Y_{ij}$ is the outcome variable for an individual $i$ residing in household $j$; $\boldsymbol{X}_{ij}$ represents the corresponding vector of observable background features such as age, education, occupation, household size, monthly household income (see Table 1); $c_j$ is the random intercept for household-level. The models used a link function, $g\left\{.\right\}$ that varied based on the type of outcome variable we used. If we used – Number of doses of the COVID-19 vaccine – as the outcome variable, then we assumed the number of vaccine shots taken follows a Poisson distribution and used a Log-link function. Hence, the model took the following functional form:

$$Pr\left(Y_{ij} = y|\boldsymbol{X}_{ij},c_j\right) = \frac{e^{-\mu_{ij}}\mu_{ij}^y}{y!} \tag{2}$$

**Table 2. Socio-demographic, economic, and other features.**

| | Duaripara N = 1239 | Tala N = 1263 |
|---|---|---|
| **Socio-demographics** | | |
| **Age** | | |
| 18–40 years | 876 (70.7%) | 681 (53.9%) |
| Above 40 | 363 (29.3%) | 582 (46.1%) |
| **Gender** | | |
| Male | 586 (47.3%) | 613 (48.5%) |
| Female | 653 (52.7%) | 650 (51.5%) |
| **Marital status** | | |
| Currently married | 1112 (89.7%) | 1154 (91.4%) |
| Others | 127 (10.3%) | 109 (8.6%) |
| **Education** | | |
| No education | 432 (34.9%) | 242 (19.2%) |
| Primary or less | 449 (36.2%) | 381 (30.2%) |
| Above primary | 358 (28.9%) | 640 (50.7%) |
| **Occupation** | | |
| Agriculture | – | 307 (24.3%) |
| Business or self-employed | 240 (19.4%) | 163 (12.9%) |
| Service | 241 (19.5%) | – |
| Day labor | 367 (29.6%) | 259 (20.5%) |
| Others | 391 (31.6%) | 534 (42.3%) |
| **Relationship with household-head** | | |
| Household-head | 579 (46.7%) | 561 (44.4%) |
| Spouse | 509 (41.1%) | 475 (37.6%) |
| Others | 151 (12.2%) | 227 (18.0%) |
| Household size [Mean (SD)] | 4.19 (1.56) | 4.12 (1.39) |
| **Household's economic features** | | |
| Monthly income [Mean (SD)] | 18737 (10083) | 14082 (10359) |
| =1 if had access to television | 619 (50.0%) | 575 (45.5%) |
| =1 if had access to smart-phone | 667 (53.8%) | 599 (47.4%) |
| =1 if member of a micro-credit program | 303 (24.5%) | 541 (42.8%) |
| **Health** | | |
| =1 if reported any chronic illness | 288 (23.2%) | 402 (31.8%) |
| =1 if ever took COVID-19 test | 125 (10.1%) | 51 (4.0%) |
| Mental wellbeing score [Mean (SD)] | 45.0 (23.1) | 48.5 (20.1) |
| **Migration** | | |
| =1 if migrated within last 12 months | 65 (5.2%) | 134 (10.6%) |
| **Geographic feature** | | |
| Distance from nearest healthcare facility [in meter; Mean (SD)] | – | 826.8 (628.7) |

where,

$$\ln(\mu_{ij}) = \beta_0 + \boldsymbol{\beta_x} \boldsymbol{X_{ij}} + c_j \qquad (3)$$

Whereas when the outcome variable was binary (e.g., taken at least two doses of the COVID-19 vaccine), we assumed it to follow a Bernoulli distribution and used a Logit-link function to estimate the coefficients. The models took the following functional form:

**Table 3. Status of COVID-19 vaccination.**

|  | Duaripara N = 1239 | Tala N = 1263 |
|---|---|---|
| Not taken any vaccine | 112 (9.0%) | 43 (3.4%) |
| One dose | 131 (10.6%) | 55 (4.4%) |
| Two doses | 823 (66.4%) | 503 (39.8%) |
| Two- and a booster-dose | 173 (14.0%) | 662 (52.4%) |

$$Pr\left(Y_{ij} = y | \boldsymbol{X}_{ij}, c_j\right) = \mu_{ij}{}^{y} \left(1 - \mu_{ij}\right)^{(1-y)} \tag{4}$$

where,

$$\ln\left(\frac{\mu_{ij}}{1 - \mu_{ij}}\right) = \beta_0 + \boldsymbol{\beta}_x \boldsymbol{X}_{ij} + c_j \tag{5}$$

For Tala, the data structure was also hierarchical representing three levels – individuals were nested in households and households were nested in villages. The models took the following functional form:

$$g\left\{E\left(Y_{ijk} | \boldsymbol{X}_{ijk}, c_{jk}, d_k\right)\right\} = \beta_0 + \boldsymbol{\beta}_x \boldsymbol{X}_{ijk} + c_{jk} + d_k \tag{6}$$

where $Y_{ijk}$ is the outcome variable for individual $i$ residing in household $j$ in village $k$; $g\{.\}$ is the link function that varied similarly as described above; $c_{jk}$ and $d_k$ are random intercepts for household- and village-level, respectively. All the random components of the equation were assumed to be normally distributed. The rest of the notations are the same as described above.

We reported Incidence Rate Ratios (IRR) for the Poisson models and Odds Ratios (OR) for the Logit models. The IRR (or the OR) for a binary or a categorical explanatory variable can be interpreted as follows: while holding all other variables in the model constant, in comparison with the reference category, the IRR (or the OR) of getting a COVID-19 vaccine shot for the explained category of $\boldsymbol{X}$ is expected to increase/decrease by a factor of the reported unit. However, we introduced the standardised values for all the continuous variables in the models. Thus, the IRR (or the OR) for a continuous variable can be interpreted as follows: while holding all other variables in the model constant, if $\boldsymbol{X}$ changes by one standard deviation, the IRR (or the OR) of getting a COVID-19 vaccine shot is expected to increase/decrease by a factor of the reported unit. All the analyses were carried out in Stata version 17.

## Spatial analysis

The descriptive spatial analysis of vaccination at the household level was carried out by mapping vaccination coverage at the household location in each of the sites. The choice of the graphical representation is defined by the semiology of graphics [26] and performed with QGIS 3.26. To examine the spatial distribution of vaccination, we used a spatial auto-correlation analysis, a global measure to determine if there is a correlation between the value of objects (vaccination) and the metric or topographic relationships between these objects. To account for neighbouring values, correlation indices are used following the Moran index [27]. It is defined as the average of the products of the normalized values of pairs of points, weighted by the distance between two points. The index took the following functional form:

$$I = \frac{N\sum_{i=1}^{n}\sum_{j=1}^{n}W_{ij}(x_i - \bar{x})(x_j - \bar{x})}{\left(\sum_{i=1}^{n}\sum_{j=1}^{n}W_{ij}\right)\sum_{i=1}^{n}(x_i - \bar{x})^2}. \tag{7}$$

where $N$ is the number of observation (points or polygons), $\bar{x}$ is the mean of the variable, $x_i$ is the value of the variable at a particular location, and $x_j$ is the value of the variable at another location, and finally, $W_{ij}$ is the weight indexing location of $i$ relative to $j$.

### Inclusivity in global research

Additional information regarding the ethical, cultural, scientific considerations specific to inclusivity in global research is included in the Supporting Information (see S2 Checklist).

### Ethical considerations

The ethical review committee of the BRAC James P Grant School of Public Health (JPGSPH) at BRAC University, Bangladesh, approved the study protocol. The reference number is IRB-19 November'20–050. Before starting any data collection activity, authorizations were collected from the respective local administrative offices of the study sites. Informed written consents were also obtained from the participants of this study. All procedures performed in this study involving human participants were in accordance with the ethical standards of JPG-SPH, BRAC University, and with the 1964 Helsinki declaration and its later amendments or comparable ethical standards. Moreover, as the survey was conducted during the COVID-19 pandemic, surveyors wore face masks and maintained a physical distance of at least 2 meters while interviewing the participants.

## Results

### Sociodemographic features

Table 2 provides an overview of the sociodemographic characteristics of the participants in Duaripara and Tala. In Duaripara, most participants (70.7%) were aged between 18 and 40 years, while the remaining 29.3% were above 40 years old. Female participants accounted for around 52.7% of the sample, and the males represented 47.3%. About 89.7% of participants were married, and nearly half (46.7%) reported themselves as household heads. The educational qualifications of the Duaripara participants varied. Approximately 34.9% did not have any formal education. However, about 36.2% had completed primary education (equivalent to five years of formal schooling), and about 28.9% obtained education beyond the primary level. The participants from Duaripara were engaged in diverse occupations. Around 29.6% were employed as day-laborers, 19.5% held service-related positions, and 19.4% were involved in various small businesses or self-employed activities. The remaining 31.6% belonged to professions that typically do not require working outside the home, such as students, retired persons, and homemakers, and we categorized them as "others". The mean monthly household income in Duaripara was about BDT. 18,737, with a standard deviation (SD) of BDT. 10,083. About 50% of participants had access to television, and about 53.8% had access to smartphones, reflecting the prevalence of technology within the Duaripara community. About 24.5% of participants reported being members of micro-credit organizations, highlighting their social capital and access to financial support. Only a tiny proportion, specifically 5.2% of the participants, reportedly migrated out of their Duaripara household within the last 12 months from the date of the survey. Around 23.2% of participants reported chronic illnesses, indicating the prevalence of long-term health conditions in Duaripara. Only 10.1%

of participants had taken a COVID-19 detection test. Our assessment also suggests that the Duaripara participants suffered from poor mental health. The mean mental wellbeing score was 45, with an SD of 23.1.

In Tala, about 53.9% of participants were 18 to 40 years old, and the remaining 46.1% were above 40 years. Around 51.5% identified as female, and the remaining 48.5% were male. About 91.4% were married, and 44.4% identified themselves as the head of their households. Regarding educational qualifications, most Tala participants (50.7%) had obtained more than primary education, and around 30.2% had completed the primary level. The remaining 19.2% had no formal education. Regarding occupation, about 24.3% were involved in agriculture, about 20.5% were day-laborers, and about 12.9% were engaged in businesses or self-employment. The remaining 42.3% were homemakers, students, retired, or unemployed. The mean monthly household income in Tala was about BDT. 14,082, with a standard deviation (SD) of BDT. 10,359. About 45.5% of participants had access to television, and about 47.4% had access to smartphones. About 42.8% of participants reported being members of micro-credit organizations. About 10.6% of the participants in Tala reportedly migrated out of their households within the last 12 months. Around 31.8% of participants reported suffering from chronic illnesses, and only 4% of participants had taken a COVID-19 test. The mean distance from the participants' household to the nearest healthcare facility was around 826.8 meters; however, this distance varied with an SD of 628.7 meters. In Tala, the mean mental wellbeing score was 48.5, with an SD of 20.1 (see Table 2).

## COVID-19 vaccine uptake

Table 3 presents the vaccination status of the participants in Duaripara and Tala. In Duaripara, approximately 9% of the participants had not received any COVID-19 vaccine at the time of the survey. About 10.6% had received only one COVID-19 vaccine shot, approximately 66.4% had received two shots, and the remaining 14% had received a booster shot along with their first two COVID-19 vaccine shots. According to the World Health Organization (WHO), full vaccination against COVID-19 is defined as receiving at least two doses. Therefore, the estimated rate of full vaccination coverage in Duaripara was approximately 80.3%.

In Tala, only about 3.4% of the participants had not received any COVID-19 vaccine. About 4.4% had received one vaccine shot, around 39.8% had received two shots, and 52.4% had received three shots (i.e., two and a booster shot). The coverage rate for full COVID-19 vaccination in Tala was estimated to be approximately 92.2% (see Table 3).

## Factors associated with the vaccine uptake

Table 4 presents the findings from regression analyses exploring the factors associated with COVID-19 vaccine uptake in Duaripara and Tala. We found that age was significantly associated with COVID-19 vaccine uptake. In Duaripara, participants over 40 had a 12% higher incidence rate of receiving the COVID-19 vaccine than younger participants (p-value = 0.04). Tala's incidence rate was 14% higher for participants aged 40 and above (p-value < 0.01). Furthermore, we explored the factors associated with attaining full vaccination and receiving a booster shot (see columns 2 and 5 and 3 and 6 of Table 4). We found that for participants aged above 40, the odds of being fully vaccinated were approximately 51% higher in Duaripara (p-value = 0.09) and about 322% higher in Tala (p-value < 0.01). Regarding booster shots, the odds of receiving a booster dose were nearly 385% higher for participants aged above 40 in Duaripara (p-value < 0.01) and about 178% higher in Tala (p-value < 0.01).

Occupation was also significantly associated with vaccine uptake. Participants involved in professions requiring less outdoor interactions, such as students, homemakers, retired, or

**Table 4. Regression results.**

| Outcome variable | Duaripara | | | Tala | | |
|---|---|---|---|---|---|---|
| | No. of doses | Taken at least two doses | Taken at least two- and a booster-dose | No. of doses | Taken at least two doses | Taken at least two- and a booster-dose |
| Family distribution | Poisson | Bernoulli | Bernoulli | Poisson | Bernoulli | Bernoulli |
| Link function | Log | Logit | Logit | Log | Logit | Logit |
| Relative risk measure | IRR | OR | OR | IRR | OR | OR |
| **Age** | | | | | | |
| 18–40 years | **Reference** | **Reference** | **Reference** | **Reference** | **Reference** | **Reference** |
| Above 40 | 1.12** | 1.51* | 4.85*** | 1.14*** | 4.22*** | 2.78*** |
| | (0.04) | (0.09) | (0.00) | (<0.01) | (<0.01) | (<0.01) |
| | 1.01–1.24 | 0.94–2.41 | 2.43–9.68 | 1.04–1.24 | 1.93–9.21 | 1.76–4.38 |
| **Gender** | | | | | | |
| Male | **Reference** | **Reference** | **Reference** | **Reference** | **Reference** | **Reference** |
| Female | 0.98 | 1.14 | 0.85 | 1.01 | 1.69 | 1.21 |
| | (0.78) | (0.67) | (0.70) | (0.91) | (0.31) | (0.57) |
| | 0.84–1.14 | 0.62–2.10 | 0.36–1.98 | 0.88–1.16 | 0.61–4.64 | 0.63–2.32 |
| **Marital status** | | | | | | |
| Others | **Reference** | **Reference** | **Reference** | **Reference** | **Reference** | **Reference** |
| Currently married | 0.91 | 0.48* | 1.25 | 0.96 | 0.58 | 0.69 |
| | (0.32) | (0.07) | (0.68) | (0.56) | (0.37) | (0.33) |
| | 0.77–1.09 | 0.22–1.05 | 0.43–3.64 | 0.82–1.11 | 0.18–1.88 | 0.33–1.45 |
| **Education** | | | | | | |
| No education | **Reference** | **Reference** | **Reference** | **Reference** | **Reference** | **Reference** |
| Primary or less | 1.01 | 0.91 | 1.22 | 1.02 | 1.42 | 0.87 |
| | (0.91) | (0.70) | (0.57) | (0.78) | (0.44) | (0.63) |
| | 0.91–1.12 | 0.58–1.44 | 0.62–2.39 | 0.91–1.13 | 0.58–3.46 | 0.49–1.53 |
| Above primary | 0.97 | 0.62* | 2.10** | 1.03 | 1.17 | 0.88 |
| | (0.57) | (0.06) | (0.04) | (0.61) | (0.72) | (0.67) |
| | 0.86–1.08 | 0.38–1.00 | 1.02–4.30 | 0.92–1.15 | 0.48–2.84 | 0.48–1.59 |
| **Occupation** | | | | | | |
| Others | **Reference** | **Reference** | **Reference** | **Reference** | **Reference** | **Reference** |
| Agriculture | | | | 1.06 | 3.59*** | 1.55* |
| | | | | (0.23) | (<0.01) | (0.08) |
| | | | | 0.96–1.17 | 1.54–8.38 | 0.95–2.53 |
| Business or self-employed | 1.14* | 2.35*** | 1.28 | 1.06 | 3.92** | 1.82* |
| | (0.07) | (0.01) | (0.56) | (0.36) | (0.02) | (0.07) |
| | 0.99–1.32 | 1.27–4.37 | 0.55–3.00 | 0.93–1.21 | 1.23–12.45 | 0.94–3.49 |
| Service | 1.21*** | 2.75*** | 4.42*** | | | |
| | (<0.01) | (<0.01) | (<0.01) | | | |
| | 1.06–1.38 | 1.59–4.76 | 2.02–9.66 | | | |
| Day labor | 1.11* | 2.01*** | 0.72 | 1.05 | 3.59** | 1.45 |
| | (0.08) | (0.01) | (0.41) | (0.43) | (0.01) | (0.24) |
| | 0.99–1.26 | 1.21–3.33 | 0.33–1.56 | 0.93–1.19 | 1.34–9.57 | 0.79–2.67 |
| **Relationship with HH head** | | | | | | |
| Household-head | **Reference** | **Reference** | **Reference** | **Reference** | **Reference** | **Reference** |
| Spouse | 1.01 | 0.80 | 1.63 | 1.00 | 0.71 | 1.04 |
| | (0.86) | (0.51) | (0.29) | (0.98) | (0.52) | (0.91) |

*(Continued)*

**Table 4.** (Continued)

| Outcome variable | Duaripara | | | Tala | | |
|---|---|---|---|---|---|---|
| | No. of doses | Taken at least two doses | Taken at least two- and a booster-dose | No. of doses | Taken at least two doses | Taken at least two- and a booster-dose |
| | 0.86–1.19 | 0.42–1.54 | 0.66–4.04 | 0.86–1.16 | 0.24–2.06 | 0.52–2.10 |
| Others | 0.88 | 0.36*** | 1.04 | 0.96 | 0.50 | 0.59 |
| | (0.14) | (0.01) | (0.93) | (0.55) | (0.18) | (0.11) |
| | 0.74–1.05 | 0.17–0.75 | 0.39–2.82 | 0.84–1.10 | 0.18–1.37 | 0.31–1.12 |
| Household size (standardized) | 0.99 | 0.99 | 0.95 | 1.01 | 1.17 | 1.03 |
| | (0.83) | (0.91) | (0.74) | (0.73) | (0.41) | (0.86) |
| | 0.95–1.04 | 0.79–1.23 | 0.68–1.32 | 0.96–1.05 | 0.80–1.71 | 0.77–1.36 |
| Monthly income (standardized) | 1.03 | 1.19 | 1.21 | 1.01 | 1.11 | 1.22 |
| | (0.30) | (0.17) | (0.29) | (0.50) | (0.58) | (0.15) |
| | 0.98–1.08 | 0.93–1.54 | 0.85–1.70 | 0.97–1.05 | 0.78–1.57 | 0.93–1.61 |
| =1 if had access to television | 1.06 | 1.23 | 1.72 | 1.04 | 1.22 | 2.09*** |
| | (0.21) | (0.29) | (0.11) | (0.26) | (0.52) | (<0.01) |
| | 0.97–1.15 | 0.84–1.80 | 0.89–3.33 | 0.97–1.13 | 0.66–2.28 | 1.26–3.45 |
| =1 if had access to smart-phone | 1.02 | 1.06 | 1.56 | 0.95 | 0.87 | 0.58** |
| | (0.66) | (0.79) | (0.20) | (0.21) | (0.67) | (0.03) |
| | 0.93–1.12 | 0.71–1.58 | 0.79–3.08 | 0.88–1.03 | 0.46–1.65 | 0.35–0.96 |
| =1 if member of a micro-credit | 0.99 | 1.06 | 0.87 | 1.05 | 1.50 | 1.50** |
| | (0.85) | (0.78) | (0.67) | (0.25) | (0.17) | (0.05) |
| | 0.90–1.09 | 0.69–1.62 | 0.45–1.66 | 0.97–1.13 | 0.84–2.69 | 1.00–2.25 |
| =1 if reported any chronic illness | 1.05 | 1.26 | 1.38 | 1.04 | 2.32** | 1.20 |
| | (0.34) | (0.31) | (0.33) | (0.36) | (0.02) | (0.39) |
| | 0.95–1.16 | 0.80–1.99 | 0.72–2.63 | 0.96–1.13 | 1.12–4.76 | 0.79–1.83 |
| =1 if ever took COVID-19 test | 1.05 | 0.92 | 1.69 | 1.01 | 0.78 | 1.10 |
| | (0.49) | (0.78) | (0.19) | (0.92) | (0.75) | (0.84) |
| | 0.92–1.20 | 0.50–1.69 | 0.77–3.73 | 0.84–1.21 | 0.18–3.47 | 0.44–2.76 |
| Mental wellbeing (standardized) | 1.01 | 0.99 | 1.22 | 1.01 | 1.17 | 1.09 |
| | (0.71) | (0.92) | (0.18) | (0.51) | (0.33) | (0.45) |
| | 0.97–1.05 | 0.82–1.19 | 0.91–1.62 | 0.97–1.06 | 0.85–1.61 | 0.87–1.37 |
| =1 if migrated in past 12 months | 0.98 | 0.94 | 1.63 | 0.87** | 0.22*** | 0.35*** |
| | (0.86) | (0.87) | (0.40) | (0.04) | (0.00) | (<0.01) |
| | 0.81–1.19 | 0.43–2.03 | 0.53–5.00 | 0.77–0.99 | 0.10–0.49 | 0.19–0.66 |
| Distance from nearest healthcare facility (standardized) | | | | 1.00 | 1.19 | 0.94 |
| | | | | (0.86) | (0.38) | (0.73) |
| | | | | 0.96–1.03 | 0.81–1.76 | 0.69–1.30 |
| Wald Chi-square statistic (p-value) | 29.2 (0.063) | 48.5 (<0.001) | 45.7 (<0.001) | 26.0 (0.166) | 41.9 (0.003) | 60.9 (<0.001) |
| No. of villages | – | – | – | 42 | 42 | 42 |
| No. of households | 625 | 625 | 625 | 596 | 596 | 596 |
| No. of observations | 1239 | 1239 | 1239 | 1263 | 1263 | 1263 |

(a) For each dependent variable, coefficient, P-value (in parentheses), and 95% confidence interval are reported in three consecutive rows, (b) asterisks indicate statistical significance (

***p < 0.01,

**p < 0.05,

*p < 0.1).

unemployed (reported as others), were less likely to receive the COVID-19 vaccine. In Duaripara, the incidence rate for receiving a COVID-19 vaccine was about 14% higher for business persons (p-value = 0.07), 21% higher for service holders (p-value < 0.01), and 11% higher for day-laborers (p-value = 0.08) (see column 1 of Table 4). However, no significant associations were found between occupation and vaccine uptake in Tala. Additionally, the likelihood of being fully vaccinated was higher for these occupations, with odds approximately 135% higher for business persons (p-value = 0.01), 175% higher for service holders (p-value < 0.01), and 101% higher for day-laborers (p-value = 0.01) (see column 2 of Table 4).

In Tala, migration was a significant factor negatively associated with COVID-19 vaccine uptake. Participants who stayed outside their household for more than 30 days in the last 12 months had a 13% lower incidence rate of receiving the vaccine (see column 4 of Table 4). Furthermore, the odds of being fully vaccinated and receiving a booster shot were approximately 78% lower (p-value < 0.01) and 65% lower (p-value < 0.01), respectively, for those who migrated (see columns 5 and 6 of Table 4).

In Tala, getting a booster dose of the COVID-19 vaccine was positively associated with access to a television, but negatively associated with access to a smartphone. Participants with access to television had 109% higher odds of receiving a booster shot (p-value < 0.01), whereas those with a smartphone had 42% lower odds (p-value = 0.03) (see column 6 of Table 4). Moreover, being a member of a micro-credit institution in Tala was associated with 50% higher odds of receiving a booster shot, indicating the positive impact of micro-credit institutions on social capital and vaccine uptake (see column 6 of Table 4).

Income and education did not significantly correlate with COVID-19 vaccine uptake, both in Duaripara and Tala. The results remained consistent when we examined regression models on full vaccination and booster shots.

To check the robustness of our findings, we also fitted models considering the number of vaccine doses as an ordinal variable and estimated the coefficients using a logit link function. We found all the primary findings robust to these different alternative estimation techniques. The results are reported in S3 Table in the Supporting Information.

### Geographic distribution

Spatial autocorrelation for Duaripara and Tala is close to zero (0.047 and 0.02), meaning that the distribution is random and does not indicate the presence of vaccination clusters. We note that vaccinated populations are present in all neighbourhoods, including those living areas prone to waterlogging. Spatial analysis thus suggests geographically uncorrelated distribution for both Duaripara and Tala.

### Discussion

By May 2023, about 85.74% of the population of Bangladesh had received two doses of the COVID-19 vaccine [28]. This remarkable progress seems to indicate that willingness to receive the COVID-19 vaccine was not a major hurdle in Bangladesh [29]. Our study in two vulnerable sites also provides evidence of Bangladesh's overall success as well as its commitment to ensuring equal access to vaccines, especially in the areas vulnerable to climate change. With values of 80.3% for Duaripara and 92.2% for Tala, overall vaccination coverage in these two areas is on par with, if not higher than, the national average, suggesting that the vulnerable communities were not forgotten during the COVID-19 vaccination campaign.

Moreover, despite being in vulnerable areas, we did not find any significant association between vaccine uptake and income, education, or place of residence. The results indicate a relatively egalitarian distribution of COVID-19 vaccination in these two vulnerable areas. The

findings are quite different from the regular immunization programs for children in Bangladesh, where household wealth and the mother's education level significantly increase the likelihood of receiving full vaccination [30].

Our study found that people over 40, who were undoubtedly at higher risk than others, were better vaccinated for COVID-19 in Bangladesh, including the booster dose. However, older people were the most hesitant at the start of the COVID-19 vaccination campaign [31]. Bangladesh's government had prioritized riskier populations (e.g., senior citizens, health professionals) when offering the COVID-19 vaccine. Our findings reflect the outcome of the strategies implemented in distributing the vaccine. Vaccination rates were also influenced by occupation. People who were involved in jobs requiring extensive outdoor interactions had a higher likelihood of getting vaccinated than those that required more indoor interactions. However, the associations were stronger for Dhaka, reflecting the strict monitoring conducted by the authorities in the capital. This brings us back to the approach of proportionate universalism [32] and the need to adapt our public health actions for different target audiences.

Moreover, we found that access to information played a significant role in receiving booster doses. Television ownership and membership in a microfinance institution were positively associated with vaccination uptake, suggesting that the promotional campaigns had reached its audiences well and social capital also helped. However, the vulnerability associated with migration was negatively associated with the likelihood of getting vaccinated in rural areas. This is one of the biggest challenges that the COVID-19 pandemic has posed to our collective and public health capacity. Managing displaced populations [33], as well as internal and external migration/mobility, is crucial in controlling current and future pandemics [34], particularly in a country dealing with high climate change vulnerabilities [22]. It would be essential to conduct further research to understand both the reasons behind these vulnerabilities and the factors that improve the likelihood of COVID-19 vaccination. However, based on the literature, we have gathered a few thoughts on the success factors of this COVID-19 vaccination program in Bangladesh.

The COVID-19 vaccination rates are at least 60% for all the ten countries in WHO's South East Asian region. Among these countries, Bangladesh, Bhutan, Nepal, and Thailand are particularly noteworthy since they have vaccinated at least 80% of their eligible population against COVID-19 [35]. However, Bangladesh's success in the COVID-19 vaccination campaign becomes more evident when we compare its vaccination rate with other low- and middle-income countries (LMICs). For example, Bangladesh and Senegal are comparable in many aspects. These two countries are relatively similar in terms of age pyramid, poverty and inequality (e.g., Gini), mortality indicators, and the challenges their health systems face due to scarce government funding. However, by September 2022, less than 10% of the population in Senegal had received the first two doses of the COVID-19 vaccine. Like other LMICs, Bangladesh has obtained international support [36] to carry out its vaccination campaigns, whereas Senegal has started receiving substantial international funding to manufacture COVID-19 vaccines.

Besides external support, willingness to receive the COVID-19 vaccine is considered one of the primary factors for low coverage. In both Senegal and Benin, the willingness to receive the COVID-19 vaccine was about 60% [37,38]. However, surprisingly in Bangladesh, the gap between willingness and the uptake of COVID-19 vaccination is completely different. An estimate in early 2021 revealed that the willingness to receive the COVID-19 vaccine was between 31% and 74.6% in Bangladesh [6,31,39]. People living in rural areas and slums (similar to our two sites) were more reluctant to be vaccinated [31]. However, the actual uptake rate surpassed the initial estimates and concerns. This discrepancy could also be attributed to the more severe COVID-19 outbreak experienced in Bangladesh compared to Senegal, where

daily new COVID-19 cases and mortalities were considerably higher during the pandemic (see S1 Fig and S2 Fig in the Supporting Information).

A part of Bangladesh's success stems from its long history of successfully implementing vaccination campaigns [40]. It helped Bangladesh develop a long tradition of expertise in building community- and partnership-based approaches, mobilising both private and public sectors, conducting door-to-door and outreach activities, and population-specific tailored awareness campaigns [12,40–43]. During the COVID-19 pandemic, Bangladesh has designed a response program using community-based approaches and was successful in managing the COVID-19 pandemic [44]. The tradition of a community-based approach also exists in Senegal; however, it has not been sufficiently mobilised [45].

In addition, Mary et al. [12] believe that the excellent collaboration between the state and civil society (e.g., NGOs) has made reaching populations living in remote areas possible in Bangladesh. These well-designed, better communicated, and efficiently managed campaigns generated positive perceptions about the vaccines among the population [46]. An online survey among the people vaccinated across Bangladesh in early 2021 shows that 85% of the population is delighted with how the campaign was carried out, and 88% would recommend vaccination to others. The survey also indicates that the vaccination procedures (e.g., registration, waiting times, volunteers to help) went smoothly, with more than half saying they waited less than 30 minutes to get their vaccine [43]. Lessons learned from the fight against COVID-19 in Bangladesh are being used to plan actions against epidemics of influenza and other respiratory pathogens [47].

This study has several limitations. First, the vaccination statuses were self-reported. We could not verify them with any records or any seroprevalence survey. This study focuses on vaccination uptake, and we have evidence of its effectiveness, even though modeling seems to confirm it [48]. Second, we targeted surveying four adults (two age groups of two majorly reported genders) from each household. Although the average household size is approximately four, we have not included all adults in our survey. Hence, the true coverage rates might be lower than our estimations. Third, we should have carried out studies on the implementation of vaccination to understand the factors that influenced it. However, our knowledge of the field and our use of reports and articles enabled us to suggest avenues for discussion. Finally, this study focuses on two fragile areas and does not claim to be generalizable for Bangladesh. Still, the findings regarding uptake are similar to national statistics.

## Conclusion

Bangladesh has a history of implementing immunization programs through effective community mobilization and health education. The credit for its high COVID-19 vaccine coverage can be partly attributed to its longstanding experience of implementing different public health interventions successfully. However, to ensure comprehensive coverage in vulnerable areas and populations, the country can focus on targeted interventions to increase uptake by addressing specific sociodemographic differences in these areas.

## Supporting information

**S1 Text. TidiER-PH narrative.**
(DOCX)

**S1 Dataset. De-identified data.**
(XLSX)

**S1 Checklist. STROBE checklist.**
(DOCX)

**S2 Checklist. Inclusivity in global research checklist.**
(DOCX)

**S1 Table. Differences in explanatory variables by vaccine status for Duaripara.**
(DOCX)

**S2 Table. Differences in explanatory variables by vaccine status for Tala.**
(DOCX)

**S3 Table. Regression results presenting outcome variable as ordinal and logit-link function.**
(DOCX)

**S1 Fig. Mean daily new COVID-19 cases in Bangladesh and Senegal.**
(TIFF)

**S2 Fig. Mean daily mortalities due to COVID-19 in Bangladesh and Senegal.**
(TIFF)

## Author contributions

**Conceptualization:** Muhammed Nazmul Islam, Manuela De Allegri, Emmanuel Bonnet, Malabika Sarker, Jean-Marc Goudet, Valéry Ridde.

**Data curation:** Muhammed Nazmul Islam, Lucas Franceschin.

**Formal analysis:** Muhammed Nazmul Islam, Manuela De Allegri, Emmanuel Bonnet.

**Funding acquisition:** Manuela De Allegri, Malabika Sarker, Valéry Ridde.

**Investigation:** Muhammed Nazmul Islam, Manuela De Allegri.

**Methodology:** Muhammed Nazmul Islam, Manuela De Allegri, Malabika Sarker, Lucas Franceschin, Valéry Ridde.

**Project administration:** Muhammed Nazmul Islam, Lucas Franceschin, Valéry Ridde.

**Supervision:** Muhammed Nazmul Islam, Valéry Ridde.

**Visualization:** Muhammed Nazmul Islam.

**Writing – original draft:** Muhammed Nazmul Islam, Valéry Ridde.

**Writing – review & editing:** Muhammed Nazmul Islam, Manuela De Allegri, Emmanuel Bonnet, Malabika Sarker, Jean-Marc Goudet, Lucas Franceschin, Valéry Ridde.

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
