## [Decision Letter · Decision Letter 0]

20 May 2024

PGPH-D-23-02525

High coverage and equitable distribution of COVID-19 vaccine uptake in two vulnerable areas in Bangladesh

Dear Dr. Islam,

Thank you for submitting your manuscript to PLOS Global Public Health. After careful consideration, we feel that it has merit but does not fully meet PLOS Global Public Health’s publication criteria as it currently stands. Therefore, we invite you to submit a revised version of the manuscript that addresses the points raised during the review process.

Please check the comments given by two reviewers. Please submit your revised manuscript by Jul 04 2024 11:59PM. If you will need more time than this to complete your revisions, please reply to this message or contact the journal office at globalpubhealth@plos.org. Please include the following items when submitting your revised manuscript:

We look forward to receiving your revised manuscript.

Kind regards,

Hsiang-Yu Yuan, Assistant Professor

Academic Editor

Journal Requirements:

1. Please include a complete copy of PLOS’ questionnaire on inclusivity in global research in your revised manuscript. Our policy for research in this area aims to improve transparency in the reporting of research performed outside of researchers’ own country or community. The policy applies to researchers who have travelled to a different country to conduct research, research with Indigenous populations or their lands, and research on cultural artefacts. The questionnaire can also be requested at the journal’s discretion for any other submissions, even if these conditions are not met.  Please find more information on the policy and a link to download a blank copy of the questionnaire here: https://journals.plos.org/globalpublichealth/s/best-practices-in-research-reporting . Please upload a completed version of your questionnaire as Supporting Information when you resubmit your manuscript.

3. We do not publish any copyright or trademark symbols that usually accompany proprietary names, eg  ©, ®, ™  (e.g. next to drug or reagent names). Please remove all instances of trademark/copyright symbols throughout the text, including ® on page 15.

4. We have noticed that you have uploaded Supporting Information files, but you have not included a list of legends. Please add a full list of legends for your Supporting Information files after the references list.

5. Some material included in your submission may be copyrighted. According to PLOS’s copyright policy, authors who use figures or other material (e.g., graphics, clipart, maps) from another author or copyright holder must demonstrate or obtain permission to publish this material under the Creative Commons Attribution 4.0 International (CC BY 4.0) License used by PLOS journals. Please closely review the details of PLOS’s copyright requirements here: PLOS Licenses and Copyright. If you need to request permissions from a copyright holder, you may use PLOS's Copyright Content Permission form.

Potential Copyright Issues:

Figs 1 & 2: please (a) provide a direct link to the base layer of the map (i.e., the country or region border shape) and ensure this is also included in the figure legend; and (b) provide a link to the terms of use / license information for the base layer image or shapefile. We cannot publish proprietary or copyrighted maps (e.g. Google Maps, Mapquest) and the terms of use for your map base layer must be compatible with our CC-BY 4.0 license. 

* Natural Earth - All maps are public domain. (http://www.naturalearthdata.com/about/terms-of-use/ )

Additional Editor Comments (if provided):

Reviewers' comments:

Reviewer's Responses to Questions

**Comments to the Author**

1. Does this manuscript meet PLOS Global Public Health’s publication criteria ? Is the manuscript technically sound, and do the data support the conclusions? The manuscript must describe methodologically and ethically rigorous research with conclusions that are appropriately drawn based on the data presented.

Reviewer #1: Partly

Reviewer #2: Partly

2. Has the statistical analysis been performed appropriately and rigorously?

Reviewer #1: No

Reviewer #2: No

3. Have the authors made all data underlying the findings in their manuscript fully available (please refer to the Data Availability Statement at the start of the manuscript PDF file)?

Reviewer #1: No

Reviewer #2: No

4. Is the manuscript presented in an intelligible fashion and written in standard English?

Reviewer #1: Yes

Reviewer #2: Yes

5. Review Comments to the Author

Reviewer #1: Please see the attached review report. I had uploaded my review report (a PDF file). Please see my review report for more details.

PGPH-D-23-02525

Title: High coverage and equitable distribution of COVID-19 vaccine uptake in two vulnerable areas in Bangladesh

Since the character count of my review report exceeds 5000 characters, I had uploaded my review as an attachment.

Reviewer #2: 1. The linear terms in the Poisson and logit models were inaccurately presented. Usually, the linear term in Poisson regression is linear only after log transformation, whereas the linear term in logit model is obtained after logit transformation. However, the linear term presented in both the models without making any transformation. As a result, the model formulation needs to be revised. Additionally, the author should verify how these changes impact the results obtained and report any changes accordingly.

2. “If we used – Number of doses of the COVID-19 vaccine – as the outcome variable, then we assumed the probability of taking a vaccine shot to follow a Poisson distribution and used a Log-link function.” Need to rethink the statement. Number of vaccine doses lie between 0 and 3 (maximum) which seems to be an ordinal form of variable. Under this assumption, some other models, i.e. ordinal logistic regression model with binomial link function, can help improve the results.

3. In model 1, the author employed multilevel generalized linear mixed effects models. However, it remains unclear which variable was treated as the random effect term. Additionally, considering urban slums like Duaripara, which typically fall under the jurisdiction of the Dhaka North City Corporation, it is important to clarify how the multilevel generalized linear mixed effects models were applied. Therefore, it is necessary to explicitly state the random effect term used in the analysis.

4. Don’t understand how the spatial analysis added extra value in the current context?

6. PLOS authors have the option to publish the peer review history of their article (what does this mean? ). If published, this will include your full peer review and any attached files.

**Do you want your identity to be public for this peer review?** For information about this choice, including consent withdrawal, please see our Privacy Policy .

Reviewer #1: No

Reviewer #2: No

While revising your submission, please upload your figure files to the Preflight Analysis and Conversion Engine (PACE) digital diagnostic tool, https://pacev2.apexcovantage.com/ . PACE helps ensure that figures meet PLOS requirements. To use PACE, you must first register as a user. Registration is free. Then, login and navigate to the UPLOAD tab, where you will find detailed instructions on how to use the tool. If you encounter any issues or have any questions when using PACE, please email PLOS at figures@plos.org. Please note that Supporting Information files do not need this step.

---

## [Decision Letter · Decision Letter 1]

30 Dec 2024

High coverage and equitable distribution of COVID-19 vaccine uptake in two vulnerable areas in Bangladesh

PGPH-D-23-02525R1

Dear Mr. Islam,

We are pleased to inform you that your manuscript 'High coverage and equitable distribution of COVID-19 vaccine uptake in two vulnerable areas in Bangladesh' has been provisionally accepted for publication in PLOS Global Public Health.

Best regards,

Julia Robinson

Executive Editor

Reviewer Comments (if any, and for reference):

Reviewer's Responses to Questions

**Comments to the Author**

1. If the authors have adequately addressed your comments raised in a previous round of review and you feel that this manuscript is now acceptable for publication, you may indicate that here to bypass the “Comments to the Author” section, enter your conflict of interest statement in the “Confidential to Editor” section, and submit your "Accept" recommendation.

Reviewer #2: All comments have been addressed

2. Does this manuscript meet PLOS Global Public Health’s publication criteria ? Is the manuscript technically sound, and do the data support the conclusions? The manuscript must describe methodologically and ethically rigorous research with conclusions that are appropriately drawn based on the data presented.

Reviewer #2: Yes

3. Has the statistical analysis been performed appropriately and rigorously?

Reviewer #2: Yes

4. Have the authors made all data underlying the findings in their manuscript fully available (please refer to the Data Availability Statement at the start of the manuscript PDF file)?

Reviewer #2: Yes

5. Is the manuscript presented in an intelligible fashion and written in standard English?

Reviewer #2: Yes

6. Review Comments to the Author

Reviewer #2: The authors responded to the comments I raised with detailed explanations and appropriately.

7. PLOS authors have the option to publish the peer review history of their article (what does this mean? ). If published, this will include your full peer review and any attached files.

**Do you want your identity to be public for this peer review?** For information about this choice, including consent withdrawal, please see our Privacy Policy .

Reviewer #2: **Yes: ** Dr. M. Pear Hossain
